Physiological and behavioral response of the Asian shore crab, Hemigrapsus sanguineus, to salinity: implications for estuarine distribution and invasion

Hudson David M. dhudson@maritimeaquarium.org dmhudson@gmail.com 1 2
Sexton D. Joseph 2 3
Wint Dinsdale 2 4
Capizzano Connor 5
Crivello Joseph F. 2
1 Department of Research and Conservation, The Maritime Aquarium at Norwalk , Norwalk , CT , United States of America
2 Department of Physiology and Neurobiology, University of Connecticut , Storrs , CT , United States of America
3 Department of Biology, Georgia State University , Atlanta , GA , United States of America
4 Momenta Pharmaceuticals , Cambridge , MA , United States of America
5 School for the Environment, University of Massachusetts at Boston , Boston , MA , United States of America
Doblin Martina
Electronic publication date: 2018 Aug 14
Publication date: 2018
Volume: 6
Electronic Location ID: e5446
Received 2018 Jan 29; Accepted 2018 Jul 25
Copyright: ©2018 Hudson et al.
Copyright year: 2018
Copyright holder: Hudson et al.
License: This is an open access article distributed under the terms of the Creative Commons Attribution License, which permits unrestricted use, distribution, reproduction and adaptation in any medium and for any purpose provided that it is properly attributed. For attribution, the original author(s), title, publication source (PeerJ) and either DOI or URL of the article must be cited.
License URL: https://creativecommons.org/licenses/by/4.0/

Keywords: Salinity tolerance, Invasiveness, Crustacean, Hemigrapsus, Survival curve, Invasive species

Funding: Schwenk Fund National Science Foundation’s Research Experiences for Undergraduates program Some resources were provided by the Schwenk Fund for completing this work, and D. Joseph Sexton was supported for the summer under the National Science Foundation’s Research Experiences for Undergraduates program. There was no additional external funding received for this study. The funders had no role in study design, data collection and analysis, decision to publish, or preparation of the manuscript.

==============================
The invasive Asian shore crab, Hemigrapsus sanguineus, is ubiquitous in the rocky intertidal zone of the western North Atlantic. A likely contributor to this colonization is that H. sanguineus is able to handle a wide range of salinities, and is thus more likely to spread through a greater geographic area of estuaries. This study investigated the salinity effects on this animal by observing survival across a range of salinities, the maintenance of hemolymph osmolality under different salinities, and behavioral preference for and avoidance of salinities. H. sanguineus showed high survival across a broad range of salinities, had little change in hemolymph osmolality over a short-term salinity shock, and behaviorally distinguished between salinities when presented with a choice, under both acclimation salinities of 5 PSU or 35 PSU. Such results suggest H. sanguineus has a hardiness for the rapid changes in salinity that happen in the intertidal zone, yet is capable of physically moving to a more optimal salinity. This enhances their competitiveness as an invader, particularly surviving lower salinities that present challenges during high-precipitation events in rocky intertidal areas, and partially explains this species’ dominance in this habitat type.

Introduction

The invasive Asian shore crab, Hemigrapsus sanguineus, is a particularly successful invasive decapod crustacean species which is now found in estuaries and open coasts in areas along the western North Atlantic and western Europe, displacing resident species (Lohrer et al., 2000; Brousseau et al., 2002; Van den Brink, Wijnhoven & McLay, 2012; Landschoff et al., 2013; Gothland et al., 2013; Gothland et al., 2014). The species has become the most abundant crab in the rocky intertidal in New England since it was first found in New Jersey in 1988 (McDermott, 1998; Williams & McDermott, 1990; Lohrer & Whitlatch, 2002; Kraemer et al., 2007; O’Connor, 2014). Previous work in this lab and by others investigated the behavioral response of the intertidal and subtidal community to this species’ presence (Epifanio, 2013; Hudson, Reagan & Crivello, 2016). Conspecific tolerance also enhances its success in overcoming resistance to invasion (Hobbs, Cobb & Thornber, 2017). However, beyond community interactions, this species’ broad salinity tolerance could be contributing to its success as an invader and for the invasiveness of the genus more broadly (Tsai & Lin, 2007; Urzúa & Urbina, 2017), so this work aimed to evaluate its survival and behavior with respect to salinity.

Salinity is of particular importance in the marine environment to delineate biotic zones in estuaries. Organisms differ along a broad spectrum in their abilities to handle salinity changes, with fishes and macroinvertebrates (including crabs) proposed to inhabit five or six biotic salinity zones (Bulger et al., 1993; Wolf et al., 2009). Salinity tolerance could therefore be used in the management of resources in the context of locational risk for invasion by a particular species.

Invasive crab species from estuarine systems often have broader salinity tolerances to withstand rapid changes in salinity common in their native locales that are predictive for their success in new systems (McGaw & Naylor, 1992b; Colnar & Landis, 2007; Fowler, Gerner & Sewell, 2011). Salinity tolerance and preference is clearly important in determining invasiveness to intertidal zones in estuaries, particularly in decapods, and sheds light on potential areas they can invade successfully. Other notable worldwide invasive decapod crustacean species that draw attention to this particular salinity tolerance character include the Chinese mitten crab, Eriocheir sinensis, which spends much of its adult life in freshwater, but its larvae require full-strength seawater to survive (Rudnick et al., 2005). In addition, the Harris mud crab, Rhithropanopeus harrisii, native to eastern North America, is found in salinities down to 0.1 PSU and is establishing itself in new areas (Reisser & Forward, 1991; Roche et al., 2009; Kotta & Ojaveer, 2012; Fowler et al., 2013). Much of the work to determine osmoregulation in crabs was initially performed on a potent worldwide intertidal/estuarine invader, the European green crab, Carcinus maenas (Towle & Kays, 1986; Cieluch, 2004). Crabs osmoregulate utilizing the posterior gill filaments (Koch, 1954; Burnett & Towle, 1990; Lucu & Towle, 2003), with far greater Na+/K+ ATPase transport proteins expressed in the posterior gill than in the anterior gill (Burnett & Towle, 1990; Koch, 1954; Neufeld, Holliday & Pritchard, 1980). This transporter’s role in osmoregulation in crabs and other crustacean species is well-established (reviewed in Lucu & Towle, 2003; Tsai & Lin, 2007). Utilizing the changes in hemolymph osmolality as a result of this transporter’s activity over time of exposure, along with behavior, can therefore be a useful determinant of the implications of salinity change in the whole animal.

H. sanguineus experiences salinity stress below 15 PSU regardless of acclimation, indicated by increased heart rate and activity level (Depledge, 1984). The congener Hemigrapsus crenulatus shows increased oxygen consumption as salinity stress increases (as salinity decreases), strong hyper regulation at low salinities, with increases in regulatory capacity as crab size increases (Urzúa & Urbina, 2017). However, this species is easily exposed to salinities below 15 PSU during a freshwater event (i.e., rain, snow) in the intertidal zone. Tsai & Lin (2007) noticed little decrease in Na+/K+ ATPase activity in H. sanguineus between 5 PSU and 35 PSU treatments, while optima studies of congeners H. crenulatus (Urbina et al., 2010) and Hemigrapsus takanoi (Shinji et al., 2009) determined a 21 PSU optimum and 24.4 PSU optimum, respectively. Similarly, our previous initial gill work found no significant change in Na+/K+ ATPase activity in posterior gill of H. sanguineus when exposed to 35 PSU, 15 PSU, or 5 PSU seawater for 7 days, but did observe a short-term increase in activity at 2 and 4 h post-treatment for 15 PSU treatments (Hudson, 2011). Therefore, H. sanguineus has similar osmoregulatory ability with this transporter regardless of treatment, but may be able to increase its activity in the short term.

Species’ responses to gradients are particularly important in determining where they will fall within a physical range (Case & Taper, 2000), so a wider tolerance will mean a wider geographic footprint is possible. Specifically, salinity tolerance levels can greatly alter distribution of species along a coast (Teal, 1958; Barnes, 1967; Engel, 1977; Felder, 1978; Young, 1978; Young, 1979; Rabalais & Cameron, 1985; Hulathduwa, Stickle & Brown, 2007; Fowler, Gerner & Sewell, 2011; D Hudson, pers. obs., 2007). Freshwater events are common in estuarine areas, and salinity can also change on an hourly timescale with the tides, meaning that organisms living there must rapidly manage these challenges behaviorally and/or physiologically. Change in salinity is one of the most common forms of stress in the intertidal zone in estuaries, and several invasive crab species are known to be euryhaline in response (Reisser & Forward, 1991; Henry, Thomason & Towle, 2006; Roche et al., 2009; Fowler, Gerner & Sewell, 2011). This underscores the value of understanding how physiological capacity is related to behavioral choice or avoidance, since both contribute to the invasiveness of a species, i.e., how well a species reproduces and extends its range from its introduction point and starts populations in new places (Rejmánek, 2011).

To investigate this interaction between physiology and behavior, the work reported here includes physiological tolerance (i.e., maintenance of hemolymph ion concentration) and survival, but also incorporates the behavioral preference of H. sanguineus as an indicator of how well they can avoid risk. Since little change was detectable in gill physiology in previous work, a behavioral approach for this work investigated sublethal effects by quantifying behavioral avoidance and hemolymph osmolality change, along with investigating differences in overall survivorship over time. This tests the idea that H. sanguineus has an ability to tolerate wide salinity changes for a significant amount of time, and can also behaviorally avoid stressful salinities at small spatial scales, as have other species (Teal, 1958; Lagerspetz & Mattila, 1961; Jansson, 1962; Thomas, Lasiak & Naylor, 1981; Ameyaw-Akumfi & Naylor, 1987; McGaw & Naylor, 1992a; McGaw & Naylor, 1992b).

Methods

Adult crabs with carapace widths between 15 mm and 34 mm were collected by hand off Avery Point in Groton, Connecticut, USA under Connecticut Department of Environmental Protection Scientific Collector’s Permits # SC-06040 and # SC-09015. Crabs were acclimated for at least 14 days in holding tanks at 35 PSU before use.

Survival

A lab-based holding study was performed to evaluate the survival of H. sanguineus immersed in a broad range of salinity treatments typical for euryhaline species. Crabs were exposed to salinity treatments of 1 PSU, 5 PSU, 10 PSU, 15 PSU, or 35 PSU for 14 days, given the observed ability of the genus Hemigrapsus to tolerate low salinities for extended periods of time (McGaw, 2001; Tsai & Lin, 2007). The 1 PSU treatment, in particular, was included to simulate the nearly freshwater surface conditions during precipitation events in estuaries and tide pools. Specimens were kept in tanks at 20 °C that corresponded with spring and fall environmental conditions from the original capture location, Long Island Sound, including a 12-hour light/dark cycle. Crabs were held in groups, and cannibalism was accounted for as a cause of mortality if it occurred upon observation of mortality events, as were molt failures. Each salinity treatment consisted of 20 males and 20 females, which were fed with shrimp pellets every day to satiety. Crab survival was monitored daily over the course of the 14-day experimental trial where dead specimens were removed.

H. sanguineus survivorship over time (i.e., the survival function) was evaluated using methods traditionally used in the context of longitudinal survival analyses. Longitudinal data provide information on the time animals either died or were last observed alive due to ongoing monitoring of survival (Cox & Oakes, 1984; Benoît et al., 2015). Such longitudinal data for H. sanguineus consist of records for each crab specimen, which include information about the occurrence and timing of an event as well as salinity treatment values and sex that might affect survival (i.e., covariates). Crabs that were still alive when last observed or at the end of the experiment were treated as “right-censored” observations, for which their time of death was unknown either because mortality did not occur or was not observed during the holding period (Singer & Willett, 2003).

A set of non- and semi-parametric longitudinal analyses were first employed to evaluate the effect of salinity treatment and sex on the H. sanguineus survival function. The semi-parametric Cox proportional-hazards regression model was initially used given its ability to simultaneously evaluate the additive effect of multiple covariates (Cox, 1972). Preliminary regression model results suggested that the survival function was only dependent upon salinity (Table S1). Consequently, the non-parametric Kaplan–Meier estimator of survival was used to preliminarily identify if each salinity treatment produced distinct survival functions (Kaplan & Meier, 1958; Cox & Oakes, 1984; Fig. S1). The Kaplan–Meier estimator follows the proportion of individuals alive as a function of time in the absence of censored observations and is well-suited for univariate analyses with multiple factor levels.

The Peto & Peto modification of the Gehan-Wilcoxon was then used to accept or reject the null hypothesis that there was no statistical difference between survival functions (Harrington & Fleming, 1982). Multiple pairwise comparisons using the Peto & Peto test with Benjamini–Hochberg corrections to adjust for significance value inflation were subsequently applied to determine if and which salinity-dependent survival functions were statistically distinct from one another. Salinity-dependent survival functions that failed to reject the null hypothesis were subsequently combined. Preliminary results indicated survival was only significantly different between the 1 PSU and the 10 PSU, 15 PSU, and 35 PSU treatments (p < 0.01), and also between 5 PSU and 35 PSU (p < 0.05) (Table S2). However, no objective procedure could be performed to combine the survival functions with confidence given inconsistencies between pairwise comparison significance values (Table S2). For instance, while survival was not statistically different between the 1 PSU and 5 PSU as well as the 5 PSU and 10 PSU groups (p > 0.05), they could not be combined since survival between the 1 PSU and 10 PSU groups was statistically significant (p < 0.01). Coarser salinity categories were therefore examined and presented for easier interpretation of results, specifically fresh (1 PSU), estuarine (5–15 PSU), and seawater (35 PSU) salinity groups.

All survival-related analyses were performed using the statistical computing software R (version 3.4.2; R Core Team, 2017) with added functionality from the associated package “survival” (version 2.38; Therneau, 2015) and “survminer” (version 0.4.0; Kassambara & Kosinski, 2017). Statistical significance was accepted at a level of p < 0.05.

Salinity preference

The behavioral preference of H. sanguineus for specific salinities was evaluated through a separate lab-based experimental trial with new specimens. This study utilized an arena that contained two 10 cm × 10 cm chambers, each with a different salinity and bubbled with an airstone, connected by an above-water bridge to offer a binary choice, consistent with past studies (Teal, 1958; Lagerspetz & Mattila, 1961; Jansson, 1962; Thomas, Lasiak & Naylor, 1981; Ameyaw-Akumfi & Naylor, 1987; McGaw & Naylor, 1992a; McGaw & Naylor, 1992b). H. sanguineus is a highly mobile crab that in initial trials actively ran back and forth between the chambers over the bridge, meaning that it was able to effectively sample the conditions of both chambers. Therefore, individual crabs could chose to either (1) stay in the initial chamber, (2) relocate to the second chamber by using the connecting bridge, or (3) remain on the bridge since these are intertidal crabs. Because this species exists in estuaries in the field and therefore along a broad salinity gradient, individual H. sanguineus were acclimated to either 5 PSU or 35 PSU for a period of 14 days prior to the experiment to test the effects of acclimation. Since these are poikilothermic animals, activity increases with temperature. As such, temperature effects on preference were quantified by acclimating specimens at either 10 °C or 20 °C at those same salinities to simulate seasonal water temperature differences and gauge the general capacity of the animals to behaviorally regulate during different seasons. An extended acclimation time of 14 days was used to account for longer exposure to lower salinities further up an estuary and during freshwater influx events, unlike the rapid changes (i.e., ∼6 h) that occur in the littoral zone. Given acclimation conditions have been shown to modify preference behavior in other crustacean species (Gross, 1957; Hernández et al., 2006), we investigated whether acclimation conditions (temperature and salinity) affected salinity preference. We determined whether specimens had any preference for a lower or higher salinity based upon the salinity and temperature during their acclimation period. Individual crabs were presented with pairwise choices between 5, 15, or 35 PSU for a period of 12 h, with final location at 12 h recorded, for 25 replicates for each sex and acclimation at two acclimation temperatures (total of ∼100 per salinity comparison). Due to the initial high activity of this crab species, the final location at 12 h was considered the “chosen” condition.

Behavioral choice of salinity data were analyzed for binary choice by chi-square test, and then the probability of leaving starting salinity was analyzed by one-way ANOVA for each of starting salinity, sex, acclimation salinity, and temperature. In order to test interactive effects between those four factors, multiple two-way ANOVAs were completed in the statistical computing software R (version 3.4.2; R Core Team, 2017).

Hemolymph response to salinity change

To quantify hemolymph osmolality response to salinity shock, crabs were acclimated to full-strength seawater salinity (32 PSU) for 14 days to normalize gene expression (Tsai & Lin, 2007), then 40 specimens were exposed for seven days (168 h) to each of the following salinity treatments: 32 PSU (control), 17.5 PSU, 10 PSU, and 5 PSU. Five animals were taken out of the 32 PSU seawater after the 14-day acclimation period and used as the initial time point for all treatments. Due to high mortality in the survival study for some of the lowest salinities (Fig. S1), the experiment was only run for 7 days. Crab hemolymph was sampled from five new animals at each post-exposure time point of 1, 2, 4, 8, 24, 48, 72, and 168 h (7 days) and frozen at −80 °C. The early time points were chosen to compare with results for other crabs, as work in another euryhaline crab, Callinectes sapidus, supports little change in observed hemolymph osmolality values within 12 h (Sommer & Mantel, 1988; Towle, 1997; Henry et al., 2002) of salinity shock. Hemolymph samples were taken with the use of a 21 gauge syringe inserted into the crab’s branchial cavity, and stored at −80 °C in 1.5 mL centrifuge tubes. Samples’ hemolymph osmolality was measured after removal from thaw, centrifuged for 1 min at 10,000 rpm, and run in duplicate on a Wescor 5100C vapor pressure osmometer. Samples were run in duplicate, with the average of the two taken as the value for that sample. These results were then analyzed by two-way ANOVA for effects of exposure time and treatment, along with interactive effects between the two. A repeated measures ANOVA would be inappropriate to analyze hemolymph data, as the individuals were sacrificed at each time point for a separate study of the upregulation of proteins in posterior gill tissue. Each time point was analyzed for differences between the four salinity treatments by a one-way ANOVA with Tukey post-hoc analysis. Time zero was left out of analysis since it was the same for all four treatments. All statistics were completed in R statistical computing software (version 3.4.2; R Core Team, 2017).

Results

Survival

The semi-parametric Cox proportional hazards regression model indicated that the H. sanguineus survival function was only dependent on salinity treatment with no effect from the sex covariate (Table S1). When survival data were grouped into broader salinity designations for ease of interpretation and applicability to representative scenarios, the non-parametric Kaplan–Meier estimator indicated that H. sanguineus survival functions for the fresh, estuarine, and seawater salinity groups were distinct (Fig. 1), which was reaffirmed by the Peto & Peto test against all three survival functions (χ2 = 26.8, d.f. = 2, p ≪ 0.001). Moreover, multiple pairwise comparisons between all broader salinity survival functions were statistically different, thereby confirming survival was distinct between groupings (Table S3). For instance, the difference in survival between the 35 PSU (highest survival) and pooled 5 PSU/10 PSU/15 PSU treatments (high survival but some mortality) was significant (p < 0.05). There were also significant survival differences between the 35 PSU treatment and the 1 PSU treatment (lowest survival rate) (p < 0.001), and between 1 PSU (lowest survival) and the pooled 5 PSU/10 PSU/ 15 PSU treatments (high survival but some mortality) (p < 0.001) (Table S3). Therefore, it seems that while there is clearly a difference between the highest and lowest salinity treatments, the middle three salinity treatments have a moderate survival rate that is significantly different from both the upper and lower salinity treatments. Interestingly, survival did not differ significantly among the three salinity groups over the first 7 days of observation (p = 0.421).

Figure 1 Survivorship over time of H. sanguineus in different salinities.

(A) The survivorship over time is depicted as the survival function (dotted line), of H. sanguineus by broad salinity categories (1 PSU = red; 5–15 [5, 10, 15] PSU = green; 35 PSU = blue). Shaded areas represent the 95% confidence bands estimated for the salinity-specific non-parametric Kaplan–Meier survival functions. Plus symbols (+) on each survival function indicate when an individual was last observed alive (i.e., right-censored). (B) The frequency table tracks the cumulative number of right-censored individuals for each survival function over time.

Salinity preference

Behavioral preference experiments indicated a significant preference (χ2 = 5.88, d.f. = 1, p < 0.05, n = 75) of H. sanguineus for 35 PSU over 5 PSU seawater at 20 °C regardless of acclimation (Fig. 2D), but no significant preference was exhibited when individuals were given a choice between 35 PSU and 15 PSU (χ2 = 0.653, d.f. = 1, p > 0.05, n = 75), nor for 5 PSU and 15 PSU PSU (χ2 = 1.174, d.f. = 1, p > 0.05, n = 69). This significance appears to come from two sources. Males at 20 °C (Fig. 2B) showed a significant preference for 35 PSU over 5 PSU (χ2 = 9.52, d.f. = 1, p < 0.01, n = 42), and also had a significant difference in preference towards 35 PSU when first acclimated to 35 PSU (χ2 = 7.2, d.f. = 1, p < 0.01, n = 20). Crabs that were acclimated to 5 PSU prior to the experiment chose 35 PSU over 5 PSU (χ2 = 4.8, d.f. = 1, p < 0.05, n = 30). Aside from these, there are no other significant effects of acclimation on final salinity choice.

Figure 2 Salinity choice at the end of 12-hour experiments.

The laboratory setup for choice is seen in (A) (photo credit David M. Hudson). The end choice after 12 h was only significantly different for the comparison of the most extreme salinities at 20 °C, which maintained significance for males (B), was not significant for females (C), and was significant overall (D).

Figure 3 Percent of H. sanguineus leaving starting salinity after 12 hours.

(A) Percent of crabs leaving the starting location as a function of starting salinity (5 PSU, 15 PSU, or 35 PSU). (B) Animals’ likelihood of leaving based on temperature of experiment, as pooled data across starting salinities. (C) Animals’ likelihood of leaving any starting salinity based on original acclimation salinity of the animal, as pooled data across all starting salinities. (D) Animals’ likelihood of leaving starting salinity analyzed by sex, as pooled data across starting salinities. All error bars are standard error.

These data were also analyzed by whether the crab left the starting salinity in any experiments (5 PSU, 15 PSU, or 35 PSU). There was a significant effect of starting salinity on whether crabs were more or less likely to leave (one-way ANOVA, F = 32.55, d.f. = 2, p ≪ 0.001, n = 635). Interactions between factors were not significant in the two-way ANOVAs used to determine interactive effects between acclimation salinity and starting salinity nor between acclimation salinity and sex (Tables S4 and S5, respectively), but the interaction between starting salinity and temperature was significant (Tables S6), and a trend exists for an interaction between sex and temperature (Table S7). Crabs that started in 5 PSU (at both temperatures), whether for 5 PSU × 35 PSU or 5 PSU × 15 PSU experiments, were more likely to leave that salinity (move to the other salinity, escape, or move onto the ramp) (Tukey’s post-hoc test, α = 0.05, p < 0.001) than those which started in 15 PSU or 35 PSU (Fig. 3A). Additionally, crabs that started in 15 PSU were more likely to leave than those in 35 PSU (Tukey’s post-hoc test, α = 0.05, p < 0.01). As the experiment was completed in both 10 °C and 20 °C (Fig. 3B), animals were 37.7% likely to leave a salinity at 20 °C, whereas at 10 °C it was 27.5% (one-way ANOVA, F = 7.475, d.f. = 1, p < 0.01, n = 635). Acclimation had a significant effect (Fig. 3C), with animals more likely to leave the starting salinity if they were acclimated to 35 PSU (one-way ANOVA, F = 7.585, d.f. = 1, p < 0.01, n = 635). There was an effect of sex on the likelihood that an animal would leave the starting salinity (Fig. 3D), with males more likely to leave at 39.9% and females leaving 25.4% of the time (one-way ANOVA, F = 15.26, d.f. = 1, p < 0.001, n = 635).

Hemolymph response to salinity change

There was a significant effect of salinity exposure (n = 160 total) on hemolymph osmolality for H. sanguineus (Fig. 4) over the course of seven days (F = 4.6371, d.f. = 7, p < 0.001), depending on salinity treatment (F = 12.0486, d.f. = 3, p ≪ 0.001), and interaction between time of exposure and treatment (F = 2.9242, d.f. = 21, p < 0.001). Salinity treatments were quite variable in hemolymph osmolality under 8 h of exposure, but did not significantly differ from one another for the 8-hour, 24-hour, and 48-hour treatments. At 72 h, hemolymph osmolality was significantly higher (one-way ANOVA, F = 7.055, d.f. = 3, p < 0.01, n = 20) in the 32 PSU treatment than both the 5 PSU (Tukey’s post-hoc test, α = 0.05, p < 0.01) and the 17.5 PSU (Tukey’s post-hoc test, α = 0.05, p < 0.05) treatments. At 168 h (7 days), hemolymph osmolality was significantly different across the four treatments (one-way ANOVA, F = 9.383, d.f. = 3, p < 0.001, n = 20) and the 5 PSU treatment was significantly lower in osmolality than all others (Tukey’s post-hoc test, α = 0.05, p < 0.01).

Figure 4 Hemolymph osmolality of H. sanguineus after 7 days of exposure to different salinities.

(A) Total 7-day exposure results of hemolymph osmolality response to the four treatment salinities, 32 PSU, 17.5 PSU, 10 PSU and 5 PSU seawater. (B) Hemolymph osmolality response across treatments within the first 24 h. Error bars are expressed in standard error, and ‘**’ is p < 0.01.

Discussion

That survival declines for H. sanguineus over time for the 1 PSU treatment (Fig. 1) is noteworthy, but even prolonged periods of freshwater influx may not be effective in keeping H. sanguineus from surviving to establish a population, since the lowest survival rate after two weeks for these animals is still 65% at 1 PSU. Maintenance of internal hemolymph osmolality over 7 days (Fig. 4) by this species is consistent with its ability to survive. The point at which mortality began to increase in the 1 PSU treatment (∼ day 5), is consistent with the significantly lower internal hemolymph osmolality for the 5 PSU treatment of the hemolymph data only after 7 days of exposure. Work with other euryhaline crabs, like Callinectes sapidus, supports that this ability to maintain hemolymph osmolality within 12 h (Sommer & Mantel, 1988; Towle, 1997; Henry et al., 2002) helps the animal deal with estuarine osmotic stress. This study of H. sanguineus observed no change in hemolymph osmolality for 48 h, underscoring the survival ability of this crab and therefore its ability to invade new areas. This finding adds to earlier work which merely indicated that stress is induced at 15 PSU seawater for H. sanguineus (Depledge, 1984).

However, survival in a particular salinity is likely different from avoidance of suboptimal salinities. H. sanguineus individuals maintain a functional amount of Na+/K+ ATPase (Tsai & Lin, 2007; Hudson, 2011) to help them navigate this constantly changing environment, and likely uses behavioral strategies to avoid suboptimal salinities. There could be a major difference with one of these physiological characters and the physiological characters of the previously dominant intertidal crab, Carcinus maenas, which still has a depressed hemolymph osmolality at 7 days in low salinity (Siebers et al., 1982; Henry, Thomason & Towle, 2006), that may have impacted its competitive interaction with H. sanguineus and facilitated the latter’s invasion. As H. sanguineus maintains internal osmolality regardless of salinity treatment over short exposures, it may be more suited than competitors to the varying conditions of the intertidal zone. Therefore, it may be able to behaviorally maintain its shelter against competitors that leave under suboptimal salinity conditions, much like its congener Hemigrapsus nudus (Corotto & Holliday, 1996; McGaw, 2001). In the littoral zone, a change in salinity can occur during each tidal cycle and during a period of prolonged precipitation or spring melting, allowing these species to maintain territory if they are not behaviorally affected.

In the behavioral salinity choice data, a true choice of salinity was a far lesser signal (Fig. 2), and less informative, than the analysis of crabs leaving the starting level of salinity (Fig. 3). The decrease in likelihood of leaving as salinity increased is expected for optimal behavioral moderation of osmotic stress, but even the level of 51.3% of crabs leaving 5 PSU after 12 h is far less than the tidal cycle. This means that a large portion of crabs would remain in intertidal areas affected by regular salinity changes. This is an important finding with respect to metabolic stressors, as crustaceans have to switch to other physiological mechanisms, notably ammonia excretion (Shinji et al., 2009; Weihrauch, Morris & Towle, 2004), in order to maintain hemolymph osmolality at low salinities. The implication of males being more likely to leave the original salinity than females, likely due to differences in overall activity level between the sexes (Fig. 3), is that males are more likely to relocate into areas that are more suitable when salinity changes, whereas females will experience greater osmotic stress. Additionally, increased frequency of crabs leaving the starting salinity with increases in temperature means that H. sanguineus will be more likely to behaviorally respond to stressful salinities at higher temperatures than at those present during winter months, perhaps resulting in some seasonal differences in osmotic stress and mortality. Although the species can strongly osmoregulate at other salinities, those individuals relocating to full-strength salinity are likely to have an energetic advantage since there is less of a need for the excretion of ammonia (Weihrauch, Morris & Towle, 2004). This is also evident by the lower frequency of crabs leaving the starting salinity if they are starting in 35 PSU. This energetics argument is clear from recent work done in the congener H. crenulatus, which showed decreasing oxygen consumption and decreasing ammonia excretion as salinity increased (Urzúa & Urbina, 2017).

A large proportion of H. sanguineus individuals stayed in stressful starting salinities of 5 PSU (48.7% of the time did not leave) and 15 PSU (69.1% of the time did not leave), indicating that H. sanguineus should maintain territory by withstanding fluctuations in salinity that happen with rain events compared to other species. Such an inter-species comparison merits further study (Lucu & Towle, 2003; Tsai & Lin, 2007). The low likelihood of moving under stressful salinities may mean that this is a common trait to the genus, like their congener H. nudus (McGaw, 2001), and could result in faster geographic expansion and increased invasiveness of multiple members of the Hemigrapsus genus by decreasing exposure to predators (Jones & Shulman, 2008). These behavioral differences may be part of what is responsible for the more subtidal distribution of C. maenas observed in previous work (Hudson, Reagan & Crivello, 2016), not seen in the intertidal zone in estuarine areas where it does not overlap with H. sanguineus (Behrens Yamada & Gillespie, 2008; Amaral et al., 2009). Decreased desiccation of the smaller H. sanguineus using microhabitats in intertidal cobble fields when compared to the larger C. maenas (Altieri et al., 2010) may also have contributed to this intertidal dominance. This is also true for the mud crab Eurypanopeus depressus in intertidal oyster reefs (Grant & McDonald, 1979).

Physiological responses will continue to be useful in models to predict future invasions and the likely finer-scale distributions and competitive interactions in a new environment (Kneib, 1984; Zacherl, Gaines & Lonhart, 2003; Kimball et al., 2004; Rudnick et al., 2005; Herborg et al., 2007). Biological invasions continue worldwide with increasing human commerce (Pimentel, Zuniga & Morrison, 2005) so predicting a species’ probable impact by utilizing behavioral along with physiological characters synthesized with ecological and biogeographical theory will help facilitate our understanding of these processes. Behavior is becoming more prevalent as an explanation for invasion success (Weis, 2010), and this study adds to our understanding of how this invader’s distribution and use pattern arises from its physiology and behavior. Combining large-scale physical models as done for H. sanguineus in the Gulf of Maine (Delaney, Edwards & Leung, 2012) with small-scale estuarine behavior will offer a far higher resolution to spatial prediction. As invasions often gain a foothold on a small scale, the overall picture must include how the species in question interacts with these parameters on the local scale, in order to more accurately predict invasion success.

Conclusions

As survival is high in this crab under low salinity conditions, freshwater input into an estuary will probably not greatly affect survival of populations of this species. The findings here indicate an advantage of H. sanguineus in surviving stressful changes in salinity during those periods, so more founding members should survive and therefore be more likely to establish in areas where it is introduced. The level at which H. sanguineus maintains its internal hemolymph osmolality, along with its high survival rate in a broad salinity range in this study, highlight its osmoregulatory character. However, as the energetic demands of this animal become more variable as temperature increases (Jungblut, 2017), it is important to investigate the interactive effect of seasonal salinity change on likely distribution.

The genus Hemigrapsus includes two prominent invaders in Europe and North America, as H. takanoi demonstrates a wide salinity tolerance (Shinji et al., 2009) and invaded Europe (originally misidentified as H. penicillatus) (Gollasch, 1998; Asakura & Watanabe, 2005) shortly followed by H. sanguineus (d’Udekem d’Acoz & Faasse, 2002) the effects of the behavioral dominance of H. sanguineus for shelter (Hudson, Reagan & Crivello, 2016) with its ability to withstand salinity changes give it a unique ability to maintain valuable intertidal shelter from predators and competitors during changes in tides and freshwater events. This in combination with broad salinity tolerance and preferences provide the opportunity for it to outlast competitors for shelter and food when exposed to suboptimal salinities.

Supplemental Information

Figure S1 Survivorship over time at each salinity treatment

The survivorship over time, or the survival function (dotted line), of H. sanguineus by salinity categories (1 PSU = red; 5 PSU = yellow; 10 PSU = green; 15 PSU = cyan; 35 PSU = blue) (top). Shaded areas represent the 95% confidence bands estimated for the salinity-specific non-parametric Kaplan–Meier survival functions. Plus signs (+) on each survival function indicate when an individual was last observed alive (i.e., right-censored). The frequency table tracks the cumulative number of right-censored individuals for each survival function over time (bottom). For greater interpretation, salinity-specific survival functions and associated 95% confidence bands are displayed individually (right).

Click here for additional data file.

Figure S2 Total percent choice results of H. sanguineus for salinities at 20 °C

These are listed by treatment: 5 PSU ×15 PSU (A: Male, B: Female, C: Total), 15 PSU ×35 PSU (D: Male, E: Female, F: Total), 5 PSU ×35 PSU (G: Male, H: Female, I: Total). Male (Left Column), Female (Middle Column), and Compiled (Right Column) data are shown, with pooled acclimation. Behavioral preference experiments indicated a significant preference (χ2 = 5.88, d.f. = 1, p < 0.05, n = 75) of H. sanguineus for 35 PSU over 5 PSU seawater at 20 °C regardless of acclimation (I), which persisted when data were analyzed by sex, as males (G) also showed this signal (χ2 = 9.52, d.f. = 1, p < 0.01, n = 42).

Click here for additional data file.

Table S1 Impact of sex and salinity on survival

Regression output coefficient table of the Cox proportional hazards regression model used to analyze the impact of salinity treatment and sex on overall survival of H. sanguineus (n = 160). Coefficient values for all categorical levels were estimated with respect to a reference level for each covariate, thus explaining the absence of the 1 PSU salinity treatment and male groups. Survival data for specimens in the 35 psu salinity group were removed from the analysis given no mortality events occurred and vastly skewed model results.

Click here for additional data file.

Table S2 Survival comparisons by individual salinities

Pairwise comparisons between five salinity treatment groups (1, 5, 10, 15, and 35 PSU) using the Peto & Peto modification of the Gehan-Wilcoxon test. Significance values were adjusted for multiple testing using the Benjamini-Hochberg procedure and bolded.

Click here for additional data file.

Table S3 Survival comparisons by salinity treatment

Pairwise comparisons between three broad salinity treatment categories (1 PSU; 5–15 [5, 10, 15] PSU; 35 PSU) using the Peto & Peto modification of the Gehan-Wilcoxon test. Significance values were adjusted for multiple testing using the Benjamini-Hochberg procedure and bolded.

Click here for additional data file.

Table S4 Two-way ANOVA table for the effects of acclimation salinity and starting salinity

Two-way ANOVA for comparison between the effects of acclimation salinity and starting salinity on frequency of those crabs leaving the starting salinity. Significant values (α < 0.05) are bolded, trends are in italics.

Click here for additional data file.

Table S5 Two-way ANOVA table for the effects of acclimation salinity and sex

Two-way ANOVA for comparison between the effects of acclimation salinity and sex on frequency of those crabs leaving the starting salinity. Significant values (α < 0.05) are bolded, trends are in italics.

Click here for additional data file.

Table S6 Two-way ANOVA table for the effects f starting salinity and temperature

Two-way ANOVA for comparison between the effects of starting salinity and temperature on the frequency of those crabs leaving the starting salinity. Significant values (α < 0.05) are bolded, trends are in italics.

Click here for additional data file.

Table S7 Two-way ANOVA for the effects of sex and temperature

Two-way ANOVA for comparison between the effects of sex and temperature on frequency of those crabs leaving the starting salinity. Significant values (α < 0.05) are bolded, trends are in italics.

Click here for additional data file.

Supplemental Information 1 Survival of Hemigrapsus sanguineus at different treatment salinities

Each data point represents the number of crabs left at a given salinity after each day, up to 14 days.

Click here for additional data file.

Supplemental Information 2 Crab binary preference experiment data for different salinity pairs

Each data point represents an individual crab’s behavior after 12 hours for the three experiment types (5 PSU ×35 PSU, 5 PSU ×15 PSU, or 15 PSU ×35 PSU), including whether it left the starting salinity, its initial acclimation salinity, the temperature of the experiment, and its sex.

Click here for additional data file.

Supplemental Information 3 Hemolymph osmolality for Hemigrapsus sanguineus after exposure to different salinities over the course of 7 days

Each data point represents the hemolymph osmolality measured for an individual H. sanguineus crab, with information about how long the animal was exposed and what salinity treatment the animal was subjected to.

Click here for additional data file.

The authors thank Stephen McCormick, Mike O’Dea, Amy Bataille and Dr. Larry Renfro for advice and equipment, Igor Gurevich and the Aneskievich lab for use of imaging equipment, and to Adriana Hudson, Kurt Schwenk, Courtney McGinnis, Mike Gilman, and Barrett Christie. Additional thanks to Hugues Benoît for his input and programming assistance with survival modeling.

Additional Information and Declarations

Competing Interests

Author Contributions

Field Study Permissions

Data Availability

The authors declare there are no competing interests. Dinsdale Wint is employed by Momenta Pharmaceuticals, Cambridge, Massachusetts and has no competing interests.

David M. Hudson conceived and designed the experiments, performed the experiments, analyzed the data, contributed reagents/materials/analysis tools, prepared figures and/or tables, authored or reviewed drafts of the paper, approved the final draft.

D. Joseph Sexton and Connor Capizzano performed the experiments, analyzed the data, authored or reviewed drafts of the paper, approved the final draft.

Connor Capizzano analyzed the data, contributed reagents/materials/analysis tools, prepared figures and/or tables, authored or reviewed drafts of the paper, approved the final draft.

Joseph F. Crivello conceived and designed the experiments, contributed reagents/materials/analysis tools, authored or reviewed drafts of the paper, approved the final draft, dr. Crivello provided physical resources and guidance on the overall design of experiments.

The following information was supplied relating to field study approvals (i.e., approving body and any reference numbers):

Field collections were approved by Connecticut Department of Environmental Protection (Scientific Collector’s Permits # SC-06040 and SC-09015).

The following information was supplied regarding data availability:

The raw data are provided in the Supplemental Files.

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
