# Peer review of "Physiological and behavioral response of the Asian shore crab, Hemigrapsus sanguineus, to salinity: implications for estuarine distribution and invasion"

_PeerJ, doi:10.7717/peerj.5446_

## Round 0.1 · original submission · Major Revisions

Based on my own reading of the manuscript and the reviewer’s comments, we recommend publication after major revision.

While the data collection has been well designed (notwithstanding reviewer 1 and 3’s comments about the life stages likely to be transported by ships), the conclusions drawn from the data and the general clarity of the writing needs improvement. One example is (line 323) “In the behavioral salinity choice data, a true choice of salinity was a far lesser signal (Fig. 2), and less informative, than the analysis of crabs leaving the starting level of salinity (Fig. 3).

Furthermore, some updating of cited literature and an explanation for why repeated measures ANOVA was not used to analyse hemolymph data is needed, as well as clarity about the modelling as outlined by reviewer 3.

Please go through your ms and revise your ms accordingly.

Reviewer 1 ·

Basic reporting

Overall, very persuasive presentation of a useful contribution to the field of invasion biology pertaining to this interesting species. There is some need for a little more basic editing for clarity and brevity. For example, the third sentence in the abstract (“We used an effective approach the…”) seems to be missing a word or two somewhere in the middle and the first sentence in the introduction would seem to benefit from some reordering to place the subject at the beginning of the sentence. I recommend just re-reading or having an outside editor go over the ms one final time to do some tightening. To improve clarity and flow, I’d also recommend placing some of the inline citations more at the end of phrases or sentences, rather than in the middle of them (see Lines 56 and 58, for example).

The literature used seems sound and supportive, though I do worry that most of the work cited is from prior to 2010, so it may not reflect more recent work.

Abstract:
Content is informative and relatively clear, given some editing. Line 39 in the abstract may need to more clearly state that you’re referring to the rocky shore habitat, rather than ballast tanks.

Introduction:
Line 53 – should you say “… crab in the *rocky* intertidal…”?
Line 56 – As much work has been done on the invasion ecology of this species, I recommend including some more sources here – at least the review by Epifanio 2013.
Line 69-70 – Unclear inclusion. Are you intending to say that C. maenas is also estuarine, and thus a useful baseline for understanding? Need to clarify.
Line 74-78 – For brevity, I’d suggest just cite the 2003 review paper and the 2007 study, rather than provide the full list of cites and species?
Line 101-102 – I’m not sure what you’re saying here, but I don’t see that FW events are any more (or less) common along the eastern US than any other coastline – at least not without citation, and not sure the basis for the spring/fall pattern assertion. I think the important detail to make is that salinity can be highly variable, especially around estuaries, and that has a direct impact on the distribution of species.
L 103-107 – this sentence seems repetitive and tautological. You make a good case that we already understand that successful coastal invaders are often euryhaline (which already means they are good osmoregulators), so it’s unclear what the “need” is that you refer to. I’d suggest rewording this to, perhaps, say express that there is still some value in understanding this common characteristic further – or, more germane to your work, say that we still need to understand the role of behavioral choice, etc…
L 110 – I’m a little unclear on how behavioral preference is an indicator of risk. Isn’t it a measure of the ability of the animal to avoid risk? I may just not quite be understanding the use of the term “indicator” in this context.
L 112 – “…during ocean crossings,” needs a citation.
L 111-113 – This sentence seems to refer to the ability of H. sanguineus (larvae?) to withstand transport in ballast, etc., yet the rest of the paragraph discuss the current project – which doesn’t seem to be looking at ocean transport or larval survival, etc. It’s an interesting and important piece of information, but I’m not sure the sentence is relevant to the paragraph or the paper.
L 118 – “maladaptive” is not the right word to use here, given its typical use in a biological context. Perhaps consider something like “stressful” instead?
L 120-122 – Is this trait truly particularly important in decapods? I’m not sure what the basis is for that claim.
Overall, this sentence reads a little rough and vague. For example, what does it mean that they can alter environments? Perhaps consider revising.
L123-124 – “…as this is an a major…” remove “an.”

Experimental design

Experimental design seems reasonably well done and explained, though there is room for some editing for content and conciseness. However, I’m worried there may be some substantial errors in thinking. Particularly, the ballast water / Euryhalinity experiment has two major flaws as currently presented: 1) it uses adult crabs, whereas ballast water transports larvae, and 2) ballast water is not typically freshwater at all, at least not as experienced by this species. See Carlton 1985 and Bailey 2015 for a start to this topic. I’d suggest that all is far from lost, though – consider simply doing away with couching this in terms of ballast and focus on the (re)testing of the tolerance of *adult* H. sanguineus to different salinities.

Methods:
L 128-130 – some more details needed here: What size / age-class crabs? What were the acclimation conditions?
L 140 – I’d suggest not calling this “room” temperature. How variable was the “~”?
L 142 – How and how much were they fed?
L 144 – “Confirmation” of death? Maybe not necessary to state.

Overall, the statistical analyses seem appropriate, though I am not familiar enough with the specifics of some of them to fully comment. I do have a little concern about the application of multiple analyses to the same data, and the seeming post hoc nature of some of the tools used. But, I do like the relatively thorough and clear description of this portion of the methods.

The behavioral choice experiment is particularly nicely done.

Validity of the findings

Results:
I appreciate the brevity with which the results are presented, but some the specific results and their stats should be more expressly stated in the body of the section. However, the data does same appropriately robust and statistically sound.

L 238 – What does the word “suggested,” lease clearly state the results of the Kaplan Meier test here. In what way do these groups differ? Please state here.
L 244 – I’m not sure which analysis did not find significance in survival. Please state and, if appropriate, give the full statistical results in addition to the p-value.

Discussion:
Generally sound conclusions based on the present study, but much of the discussion is speculative and only loosely based on this and prior work. Again, I also strongly suggest some overall editing for flow and conciseness.
L313-315 – this is interesting speculation, but not clearly sound based on the current study. Please state more clearly.
L 332-334 – sentence unclear. Please reword.
L342 – “…may make it able to use of microhabitats…” missing word.
L343 – “…lesser…” should probably read “less.”

Additional comments

I think this paper makes a useful contribution to our overall understanding of this particular invader crab, and the role of salinity stress on shaping distribution. Given some reworking of some concepts (particularly regarding ballast water), a little clarification on some analyses, and some substantial style editing, this paper will be very worth of publication in PeerJ.

I look forward to seeing this work published.

Reviewer 2 ·

Basic reporting

I feel there a bunch of areas where the grammar/descriptions can be tightened up or clarified. For instance:

In the abstract, "running trials" sounds like it could be "shrimp on a treadmill" or the process of conducting multiple trials.

line add "to" or something, e.g. "approach TO the salinity"

Line 36 I argue that "willingness" of crabs cannot be evaluated. Perhaps the authors could change to "capable"

Lines 57-60 can be shortened substantially.

Lines 74-81. I don't think ALL the species need to be listed. There are even some repeats. Perhaps just the authors.

Line 84 can be shortened to "H. sanguineus experiences salinity stress below 15 PSU..." "seawater" seems redundant.

Lines 108-111 can be shortened.

Lines 116-123. "This" is often vague, and it is used a bunch here. Also, I would prefer to see and end to the introduction dealing with the specifics of the study, not general statements about invasiveness. The salinity-invasiveness relationship should be mentioned earlier in the intro.

Lines 135-139 should be rearranged so grammar and logic are more inline.

Lines 146-149 are largely unnecessary.

Lines 153-164 - Right-sensored is not clear. Is it possible to briefly explain what this means? Are other references to sensored data then "left sensored"?

Lines 183-184 Beginning with "only two salinities" sounds like it is insufficient in some what. Do you mean that there is a convention of using 2 salinities? and you follow this convention?

Experimental design

I don't understand why the results in Figure 3 are all reported separately. It sounds like the experiment is a factorial design with male/female exposed to the different salinities and temps, but the data just show individual bar graphs of single factors. More importantly the analysis should look at interactions between factors if >1 factor was in an experiment. Some of these analytic and design details are obscured by the often vague description in the methods.

Whys does table S1 only report female? This could be clarified in the text below S1.

Lines 153 - how many time of deaths were unknown?

Validity of the findings

Most of the discussion focuses on the osmolality across salinities, but most of the Methods/Results is about sex and acclimation differences. Can these be brought more inline?

Reviewer 3 ·

Basic reporting

1) There a few places in the manuscript where statements are made without any supporting references. Appropriate citations need to be added. I have added “notes” to the PDF indicating these (ex. lines 112, 139, 334, and others noted in text).
2) There are also several places in the manuscript where the meaning of certain sentences or phrases is unclear. I have added “notes” to the PDF to point these out.
3) The context for the study developed in the introduction and discussion focuses on the implications of salinity tolerance for promoting or preventing dispersal of organisms via ballast. While the tolerance of adult crabs to salinity variation is highly informative for distribution potential within the invaded habitat, a more appropriate test of salinity tolerance would have to focus on larvae instead of adult crabs. Because this study is focused on adult crabs, the authors should address the potential correlation between adult and juvenile/larval tolerance (using examples from literature), and should reduce the focus of the interpretation on this particular point. Instead, the focus could shift to the role of salinity tolerance in the likelihood of distribution to or occurrence in different habitat types.

Experimental design

The number of samples used in experiments is quite good. There are few areas where descriptions of methods should be elaborated or clarified:

4) For the survival experiment, were crabs held individually or in groups? If in groups, how was cannibalism (common in this species) prevented or accounted for as a cause of mortality for the survivorship study?
5) The number of treatments that are reported is collapsed from the original 5 treatments into just 3 categories. The reason for changing the analysis and reporting seems to be (from wording at lines 171-173) that the analyses of each independent treatment did not show significant differences between survival from on treatment to another, so they were collapsed. Is this interpretation correct, and if not, could this section be re-explained?
6) Line 157-159 – what were the original set of parameters that were included in the models, before the final dependent variables were identified (ie. What was included in the models besides salinity?)
7) The methods for the behavioral choice experiment do no include enough detail to be replicated. The details of the trails themselves (how long a crab was monitored, how long it had to remain in its chamber or remain in the other chamber in order to be considered to have “chosen” those conditions) need to be explained.
8) While the setup of the behavioral trials might allow for compelling evidence of the likelihood of movement out of the starting container, the approach does not appear to provide a signal allowing crabs to detect options outside of the chamber they are started in (as would be the case with a y-tube set-up). This should be discussed.
9) Lines 192-193 – this is the first time that a potential effect of temperature on salinity tolerance/avoidance is brought up. A bit more context could be given here or in the introduction to explain why multiple temperature treatments are used.
10) Lines 194-202 – How do temperatures of the behavioral choice containers compare to acclimation temperatures? Were these set up so that crabs acclimated at 10 or 20C were in arenas that corresponded to these temperatures during the experiment? The temperature set-up is no currently clear. Line 201 also says salinity comparisons were varied between 5,15, and 30 – this sounds like salinities were changed? The wording of this sentence should be changed for clearer interpretation.

Validity of the findings

These comments pertain particularly to the interpretation/discussion of the reported results:
11) The while the first paragraphs of discussion do a good job of getting into the implications of the findings, there is little discussion or interpretation or several of the experimental results reported. In particular, there is no discussion of the results of the behavioral assay: males more likely to leave original container than females; effect of temperature on likelihood of leaving; preference for 35 PSU; effect of acclimation salinity on likelihood of leaving container.
12) The discussion also makes several statements about the relative competitive ability of H. sanguineus compared to other intertidal crabs, in terms of the possible advantage that H. sanguienus may gain due to their physiological salinity tolerance. However, none of these statements are supported by comparisons to existing data about the salinity tolerances of other species with which H. sanguineus may interact. I have listed a few relevant references below that the authors may wish to explore, in order to better inform (and potentially support) these statements.
13) Lines 338-348 that expand on ideas of how H. sanguineus may exclude other crab species from the intertidal is interesting, but getting pretty unrelated to the results of the present study. This information should be pared down in favor of adding additional discussion of the experimental results that are currently missing.
14) The final couple of paragraphs of the discussion (lines 349-376) are quite broad, and provide some information that might be very helpful to cover in the introduction rather than the discussion. For example, lines 363-366 succinctly introduce the idea that species vary in their salinity tolerances, and that these differences can structure species in space in estuaries. This kind of information seems better suited to the introduction, perhaps in the first paragraph.

Additional comments

This study explores the effect of salinity acclimation on mortality and hemolymph osmolality of Hemigrapsus sanguineus, a non-native shore crab that has become highly abundant on rocky shores throughout the Western Atlantic from Chesapeake to southern Maine. The authors also explore whether acclimation to different salinities influence the likelihood of crabs exiting their holding containers and whether the crabs end up in containers with different starting salinity. The results of the study are interpreted in terms of how salinity tolerance may affect likelihood of H. sanguineus outcompeting other crab species, and surviving transit in ballast tanks.
The questions addressed in this study (whether H. sanguineus is behaviorally and physiologically adaptable in a way that may promote invasion success) is important, and the methods and level of replication are well-done. The majority of my suggestions relate to the interpretation of the data collected, and to the context and clarity of its presentation. I have provided general comments related to these concerns, and I have added additional notes directly to the review PDF with suggestions to improve clarity of the writing. Overall, this is a meritorious study that will contribute to our ability to predict the current and potential distribution of this crab species.

Annotated reviews are not available for download in order to protect the identity of reviewers who chose to remain anonymous.

---

## Round 0.2 · Minor Revisions

You have responded well to the reviewer's queries about your original submission - thank you - but a few remaining clarifications are needed. I have inserted a few minor editorial comments of my own in the attached PDF.

Reviewer 3 ·

Basic reporting

The overall clarity of the manuscript has been significantly improved, and I appreciate the authors attention to detail in addressing comments and suggested edits provided in the previous review. There are a few remaining or newly introduced sentences that are slightly confusing, and I have added comments directly to the .doc version of the manuscript to point these out (apologies, but line comments were not possible as the line numbers re-start on each page, and no page numbers were included - something the authors may wish to attend to for future submissions).

The context for the paper has also been improved, with the new emphasis on the implications of wide salinity tolerance for geographic spread of established populations.

Experimental design

The clarity of methods reporting has also been much improved, including reporting of analyses used. I have added a few additional comments on the .doc where ambiguities remain.

Validity of the findings

The reporting of results is also much clearer, and I appreciate the authors' more detailed discussion of the implications of the results. A few minor comments are included in the manuscript.

Additional comments

The revised version of this manuscript coherently reports on new and sound data pertaining to the behavioral and physiological osmoregulatory capacity of an important invasive crab species. The organization and clarity of this version are good, and the paper and should be considered for publication after correction of a few minor edits that will further clarify and streamline the text.

Annotated reviews are not available for download in order to protect the identity of reviewers who chose to remain anonymous.

---

## Round 0.3 · Minor Revisions

While your submission is much improved, there are a number of edits and comments that remain in your updated manuscript - I have attached a marked up PDF.

In addition, please review all of your supplementary tables and use three decimal places when reporting p values.

---

## Round 0.4 · accepted · Accept

I have made a few minor edits to your text (e.g. in the Introduction, 'Salinity tolerance could therefore be used in the management of resources in the context of locational risk for invasion by a particular species.'; and in the Conclusion, replaced 'maladaptive' with suboptimal salinity). Your annotated manuscript is attached.

#